# Validation of reference genes for RT-qPCR relative expression analysis during cyst-to-early adult development of *Taenia solium*

Jose Maravi©, David Castaneda-Carpio©, Renzo Gutierrez-Loli©¤a*, Segundo W. Del Aguila¤b, Valeria Villar-Davila, Juan Blume-La-Torre¤c, Cristina Guerra-Giraldez*

Laboratorio de Proliferación celular y Regeneración, Laboratorios de Investigación y Desarrollo, Facultad de Ciencias e Ingeniería. Universidad Peruana Cayetano Heredia, Lima, Peru

© These authors contributed equally to this work.
¤a Current address: Toulouse Institute for Infectious and Inflammatory Diseases, Université de Toulouse, CNRS, Inserm. Toulouse, France
¤b Current address: Villanueva Lab, Molecular and Cellular Oncogenesis Program, The Wistar Institute. Philadelphia, Pennsylvania, United States of America
¤c Current address: Department of Biological Chemistry, University of Michigan Medical School. Ann Arbor, Michigan, United States of America
* renzo.gutierrez@upch.pe (RG-L); cristina.guerra@upch.pe (CG-G)

## Abstract

The development of the zoonotic parasitic tapeworm *Taenia solium* from larval to adult involves significant but often clinically overlooked events crucial in cestode biology. The early-adult events can be studied *in vitro*, providing a valuable model to examine scolex evagination, strobilation, and worm development. With some transcriptomes being reported, single-gene relative expression analysis using reverse transcription of RNA (RT) followed by quantitative PCR (qPCR) is valuable to confirm differential expression and study gene regulation during parasite development. However, accurate comparisons with this approach require the validation of endogenous reference genes (RGs). This study identifies stable RGs for normalizing transcript expression data in *Taenia*. We examined 12 candidate RGs across three "early tapeworm" phases grown in culture. Transcripts were evaluated with RNA-seq and qPCR. Stability rankings were generated using *geNorm* and *NormFinder* (for RNA-seq) and *RefFinder* (for qPCR). Transcripts for *rpl13* and *ef1a* were ranked as the most stable and were tested by using them to normalize the expression of *h2b* and *wnt11a*, involved in proliferation and strobilation processes.

## Author summary

*Taenia solium* is a parasitic tapeworm that causes taeniasis and cysticercosis, diseases that severely affect populations in low-resource regions. While the cystic larval stage in pigs is well characterized, much less is known about the

**Data availability statement:** The work includes analyses that build upon the dataset we have made available in the Gene Expression Omnibus (GSE288552), published as a Data Descriptor in Scientific Data (doi.org/10.1038/s41597-025-05141-2). Other relevant data are within the manuscript and its Supporting Information files.

**Funding:** JM, DCC and VVD were supported by grant PE501079376-2022 from Consejo Nacional de Ciencia, Tecnología e Innovación tecnológica, CONCYTEC, through its Peruvian Programa Nacional de Investigación Científica y Estudios Avanzados, Prociencia (https://prociencia.gob.pe) awarded to RGL and CGG, which also funded the completion of this work. Besides encouraging publications. The funders had no role in study design, data collection and analysis, decision to publish, or preparation of the manuscript.

**Competing interests:** The authors have declared that no competing interests exist.

parasite's transformation into the adult tapeworm, which occurs exclusively in the human intestine. This process is critical for transmission, as adult tapeworms release eggs that spread infection. In this study, we used cysts in culture representing the early events of adult worm development. Our goal was to identify stable reference genes necessary for accurately measuring gene expression during this process. Building on our previously generated *in vitro* gene expression dataset focused on initial adult development, we validated key reference genes that will support future research into parasite biology and may contribute to strategies aimed at controlling its spread and reducing disease burden.

## Introduction

The genus *Taenia* includes over a hundred species of tapeworms, parasites of wildlife and livestock. Three among these, *T. solium*, *T. saginata*, and *T. asiatica,* cause zoonotic infections, leading to taeniasis and cysticercosis. These helminthiases are part of the neglected tropical diseases prioritized by the WHO [1].

Cysticercosis can cause life-threatening conditions, particularly in poor, remote areas of developing countries in Africa, Asia, and Latin America, where it impacts the health and livelihoods of rural farming communities. A 2015 WHO report identified *T. solium* as a leading cause of death from food-borne diseases, resulting in a total loss of 2.8 million disability-adjusted life-years (DALYs) [2,3].

*T. solium* has a complex life cycle involving pigs as intermediate hosts (cysticercosis) and humans as definitive hosts (taeniasis). Individuals who carry an intestinal tapeworm shed microscopic embryonated eggs, which—upon ingestion by pigs or humans—invade tissues and develop into cysts [4]. Poorly cooked pork can carry viable cysts that, once activated in the human gut by gastric and bile acids, evaginate the scolex to attach to the intestinal wall, and grow and mature into an adult tapeworm [4,5].

Although the complete life cycle of *T. solium* has yet to be reproduced in the laboratory, studies in animals have yielded valuable knowledge about the egg-to-cyst (larval) development [6,7]. There is significantly less information available on cyst-to-adult development, partly because it occurs within the human host and because taeniasis typically presents with mild or non-specific symptoms [4,8]. Understanding this process is essential, as human carriers are key drivers of transmission and endemicity [9,10].

*T. solium* cyst-to-adult development involves key processes like scolex's evagination and further strobilation [11], which can be addressed *in vitro*, starting from chemically [12] or enzymatically treated cysts [13]. We have recently made available a bulk RNA-seq transcriptome from *T. solium* cysts collected during *in vitro* activation and early growth [14]. Complementing this and other datasets [15–17], RT-qPCR single-gene analysis remains valuable for validating differential expression during development, focusing on specific genes, and comparing expression across tissues. Proper normalization via stable RGs (e.g., *gapdh,* for glyceraldehyde-3-phosphate

dehydrogenase) [18] is essential [19]. A few analyses of expression stability in RGs have been conducted in other cestodes [20,21], but a similar validation for the pre-adult phases of *T. solium* is lacking, as well as defined stages characterized by anatomical milestones, as has been done for the cestode laboratory model *Hymenolepis* [22].

This study identifies RGs for analyzing gene expression in *T. solium* during *in vitro* cyst activation and development. The stability of twelve candidate RGs was analyzed using RNASeq and qPCR data, and the top-ranked genes were used to normalize the expression of two transcripts that likely show differential expression during scolex evagination: *h2b*, related to cell proliferation [23], and *wnt11a*, involved in posterior development and strobilation [24].

## Methods

### Ethics statement

The study protocol was revised and approved by the Animal Care and Use Committee (CIEA, for Comité Institucional de Ética para el Uso de Animales) at Universidad Peruana Cayetano Heredia, which states details for the proper housing, feeding, and handling of animals used in research, including euthanasia. The CIEA is registered with the Office of Laboratory Animal Welfare of the Department of Health and Human Services, National Institutes of Health (NIH–USA), under Assurance Number F16-00076 (A5146-01), valid until August 31, 2026.

An overview of the study design and analysis workflow is seen in Fig 1. We selected 12 candidate RGs, and after identifying sequences for their orthologs in the *T. solium* genome (S1 Table), we evaluated the expression stability of the corresponding transcripts using two complementary approaches. First, we analyzed RNA-seq data from our *T. solium* transcriptome, which covers three morphologically distinct phases surrounding *in vitro* scolex evagination in the presence or absence of taurocholic acid (TA) as an evagination inducer [14]. We then performed two-step RT-qPCR on cysts obtained from a different culture under identical conditions and sampling points. This design enabled both *in silico* and experimental evaluation of RG stability across biologically relevant phases. Validation of the candidates as RGs was done by using them to normalize qPCR results of target genes *h2b* and *wnt11a*.

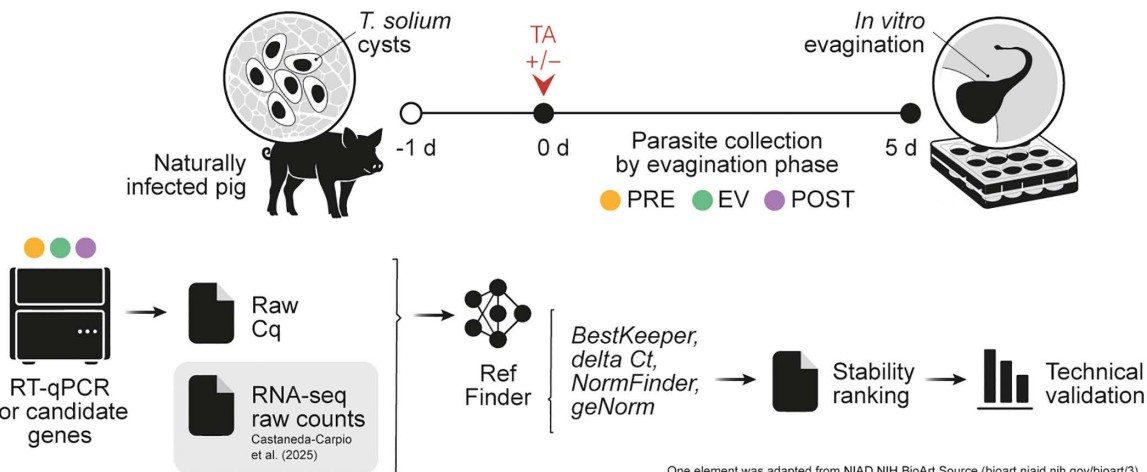

**Fig 1. Study design.** The stability of candidate RGs across three distinct phases in the *in vitro* activation and early growth of *T. solium* cysts was evaluated using a dual approach combining RNA-seq and RT-qPCR data from independent experiments. The most stable candidates, as determined by both methods, were technically validated. Figure created by the authors; some elements adapted from NIAID NIH BioArt Source (bioart.niaid.nih.gov/bioart/3).

## Selection of candidate RGs

We selected 12 candidate RGs, listed in Table 1, based on their use in qPCR studies focused on invertebrates and their involvement in essential cellular functions such as transcriptional regulation, protein synthesis, cytoskeletal structure, energy metabolism, and mitochondrial activity. These genes are evolutionarily conserved and may exhibit stable expression across varying conditions, making them suitable candidates for normalization in RT-qPCR analyses.

## RNA-seq data analysis of transcript expression stability

The stability of candidate reference genes was assessed using the *T. solium* transcriptome dataset, available in the Gene Expression Omnibus (GSE288552) [14]. Counts per million (CPM) values were calculated with edgeR [31] from RNAseq raw counts. Transcript stability of the candidate RGs, except *cox1*, which was not part of the transcriptome, was calculated using CPM-normalized input for *geNorm* [32] and $\log_2(\text{CPM} + 1)$ for *NormFinder* [33]. We applied the recommended *geNorm* M-value threshold of 1.5, where lower values denote greater stability; similarly, a cutoff of 0.15 was used for *NormFinder*, with decreasing values indicating increasingly stable expression.

## Parasite collection and culture

All procedures for collecting *T. solium* cysts from pig muscle tissue under aseptic conditions and for culturing them were as described [14]. Aside from the 72 parasites used for the reported transcriptome, an additional sixty parasites were used for qPCR analysis following the same culture conditions, including the addition of 0.1% TA (Sigma, St. Louis, MO) to the medium of half of the cysts (30 TA− and 30 TA+). The three morphologically distinct phases were also as described, namely cysts with an invaginated scolex (PRE), cysts with early scolex exposure (EV), and cysts showing clear proglottids (POST) [14]. We performed three independent experiments using separate cultures to confirm that taurocholic acid (TA) induces scolex evagination. These experiments included 60, 120, and 48 cysts, respectively, with each group evenly divided between TA− and TA+ conditions. The percentage of evagination, defined as the proportion of parasites with a visibly evaginated scolex, was recorded in each experiment, and TA− was compared to TA+. Statistical analysis was performed using two-way ANOVA followed by Tukey's multiple comparison test.

**Table 1. Candidate RGs selected for expression stability analysis in *T. solium* cysts.** Summary of candidate RGs considered, including associated biological processes, transcript names, source organisms, and supporting references.

| Process | Transcript | Organism | References |
|---|---|---|---|
| Gene transcription | TATA-box binding protein (*tbp*) | *Echinococcus granulosus* *Phaedon brassicae* | Espínola *et al.* [21] Ma *et al.* [25] |
| Translation | L13 ribosomal protein (*rpl13*) | *Echinococcus granulosus* | Espínola *et al.* [21] |
| | Elongation factor 1 alpha (*ef1a*) | *Echinococcus granulosus* | Espínola *et al.* [21] |
| Protein folding and quality control | Ubiquitin conjugating enzyme E2 (*ube2*) | *Rapana venosa* | Song *et al.* [26] |
| Signal transduction | Mitogen activated protein kinase 3 (*mapk3*) | *Echinococcus granulosus* | Espínola *et al.* [21] |
| Cytoskeleton and cell structure | Beta actin (*βact1*) (*βact2*) | *Echinococcus granulosus* | Espínola *et al.* [21] |
| Energy metabolism | Glyceraldehyde-3-phosphate dehydrogenase (*gapdh1*) (*gapdh2*) | *Taenia solium* *Echinococcus granulosus* *Mesocestoides vogae* *Spirometra mansoni* | Orrego *et al.* [17], Hou *et al.* [18], Espínola *et al.* [21], Hayashi *et al.* [27], Wang *et al.* [28] |
| | Phosphoglycerate kinase 1 (*pgk1*) | *Mesocestoides vogae* | Hayashi *et al.* [27] |
| | Malate dehydrogenase (*mdh*) | *Meloidogyne hapla* | Wu *et al.* [29] |
| Mitochondrial respiration | Cytochrome *c* oxidase subunit I (*cox1*) | *Penaeus monodon* | Hembrom *et al.* [30] |

## Primer design for qPCR

Primers were designed using Primer3 [34] and BLAST software based on the mRNA sequences derived from *T. solium* orthologs for the selected candidate RGs and target genes, retrieved from WormBase ParaSite databases [35] (S1 Table). Version 1 of the *T. solium* genome was used; orthologs were identified based on sequence identity and domain conservation, and validated with BLAST. The basic parameters used for primer design were as follows: primer length between 18–21 bp, GC content between 45–55%, melting temperature between 55–64°C, and amplicon length between 60–200 bp. For most genes, primers were located within single exons (S1 Table).

## Amplification efficiency and analytical performance

To determine the PCR amplification efficiency for every candidate RG, 5-point calibration curves of cDNA (generated as described in the next section) were prepared using 2-fold, 5-fold, or 10-fold serial dilutions. The linear regression slopes and *y*-axis intercepts for every standard curve were calculated using GraphPad Prism software version 6.01 (GraphPad, San Diego, CA). Every dilution was tested in triplicate, and amplification efficiency was determined based on the slope of the log-linear portion of the calibration curve ($10^{-1/slope} - 1$), following the Minimum Information for Publication of Quantitative Real-Time PCR Experiments (MIQE) guidelines [36].

## Two-step RT-qPCR

Ten parasites per group (PRE, EV and POST) and culture condition (TA– and TA+) were used, sixty in total. As previously described, RNA from each parasite was extracted using the Quick-RNA Miniprep Plus Kit (Zymo, R1058) and treated with DNase I to prevent contamination with genomic DNA; integrity was assessed by 1% bleach agarose gel electrophoresis [14]. cDNA for qPCR was generated by reverse transcription from 100 ng of RNA of each parasite with the High-Capacity cDNA Reverse Transcription Kit (Thermofisher Scientific, Waltham, MA) in a final volume of 20 µL. The reaction mix was incubated for 15 min at 25°C, 2 h at 37°C, and 5 min at 85°C, and stored at -20°C until required. qPCR reactions (three technical replicates per sample) were done with the PowerUp SYBR Green Master Mix (Thermofisher Scientific, Waltham, MA). For every 10 µL reaction, forward and reverse primers were added at a final concentration of 500 nM each, and 1 µL of cDNA was used as the template (5 ng). The cycling conditions were set in the C1000 Dx Thermal Cycler with CFX96 Real-Time System (1841000-IVD, BIO-RAD) as follows: 2 min at 50°C, 2 min at 95°C; and 40 cycles of 15 s at 95°C and 1 min at 60°C. Every sample was run in triplicate, and a melting curve step was done at the end of every experiment, ranging from 60°C to 95°C with increments of 0.2°C. To determine the quantification cycle (Cq) for each sample, a relative fluorescence unit (RFU) threshold for qPCR amplification was manually set for each transcript. This was based on the average of the automatic thresholds generated by the Bio-Rad CFX Manager 2.1 software for each plate run, ensuring consistency across all samples. Care was taken that all average thresholds fell within the linear segment of amplification curves. RG datasets with Cq values greater than 35 were excluded due to low transcript abundance. The size and specificity of the amplification products were verified by 1.5% agarose gel electrophoresis.

## Stability analysis using RT-qPCR data

The RT-qPCR Cq values of each RG were compared and ranked with different normalization algorithms integrated in the free web-based tool *RefFinder* [37], available at: https://www.ciidirsinaloa.com.mx/RefFinder-master/ [38] using raw Cq values as input to provide a geometric mean ranking of RG stability. This program integrates the analysis performed by the extensively used computational algorithms *geNorm* [32], *Normfinder* [33], *BestKeeper* [39], and the comparative *delta Ct* method [40]. The selection of the most stable RG was based on the recommended comprehensive ranking calculated by the geometric mean of the stability values obtained from each algorithm [37]. For *NormFinder*, *geNorm,* and the *delta Ct* method, lower stability values indicate more stable expression, while for *BestKeeper,* lower standard deviation of the crossing points indicates higher stability.

## Validation of candidate RGs

To validate the reliability of the selected RGs, the relative expression of *h2b* and *wnt11a* was normalized using the geometric mean of the two most stably expressed reference genes across all conditions. Both *h2b* and *wnt11a* have been reported as differentially expressed upon scolex evagination in the parasitic tapeworm model, *Hymenolepis sp.* [41,42]. The $\log_2$(CPM + 1) values of *h2b* and *wnt11a*, obtained from the *T. solium* transcriptome [14], were compared across PRE, EV, and POST, under TA+ and TA− conditions. The statistical *t*-test was applied; significant differences are indicated with $p < 0.05$. Ten individuals per condition and three technical replicates per sample were set up for qPCR analysis. Fold-change for *h2b* and *wnt11a* at experimental conditions was analyzed using the $2^{-\Delta\Delta Cq}$ normalization method [43] and the geometric mean of the two genes with stably expressed RG across all conditions using the Pfaffl method. The sequence and other technical details for *h2b* and *wnt11a* primer sets are shown in Table 2.

## Results

### *In vitro* parasite evagination

Direct observations, along with quantification of cysts with evaginated scolices from three previous experiments, demonstrated that from day 2 onward, cysts cultured in the presence of 0.1% TA consistently exhibited twice the evagination rate

**Table 2. Primer sequences against *T. solium* transcripts for the 12 RGs and two target transcripts, and RT-qPCR amplification features.**

| Transcript | WormBase ParaSite accession no. | Primer sequence 5'→3' (forward/ reverse) | Product size (bp) | T$_m$ (°C) | %E |
|---|---|---|---|---|---|
| *bact1* | TsM_001199400 | ATCGTAGCACCACCAGAACG/ CCCGATTCGTCGTACTCCTG | 113 | 60.0 | 94.13 |
| *bact2* | TsM_001001400 | GTGCGAGACATCAAGGAGAAG/ GGAAGCGTTCATTGCCAACT | 142 | 60.4 | 98.31 |
| *gapdh1* | TsM_000056400 | TCCAAGAGATGAATGCCAATGC/ CAGAAGGAGCCGAGATGATGA | 147 | 61.1 | 93.89 |
| *gapdh2* | TsM_001083000 | CAAACTTGCCAAACCTGCCA/ GCGTCGAAAGTGGATGATGC | 153 | 58.7 | 97.87 |
| *pgk1* | TsM_000796500 | GACACCGCTACTTGTTGTGC/ TCCGAAAGAGCTGATACGCC | 125 | 59.0 | 105.09 |
| *mdh* | TsM_000048200 | TAAGGTGCTGGTTGTTGGCA/ CGACCTGATAGATGGCACGG | 131 | 59.8 | 91.18 |
| *cox1* | TsM_000113400 | CCGTTAGGAGGTGGTGATCC/ ACCCATAAAAGCCAAAAGCA | 154 | 60.1 | 101.69 |
| *tbp* | TsM_000071900 | GCCAAAGTGCGAGATGAGAT/ CCCGATTTGTCTGAATCCAA | 89 | 56.8 | 96.37 |
| *rpl13* | TsM_000621100 | TCCAATGCTATCGGGTGAAT/ CGCCTCGTGGATCGTATATT | 74 | 56.0 | 103.09 |
| *ef1α* | TsM_000233800 | TTGCTGCTGGTACTGGTGAG/ CGCTATAATCAACCGCATCC | 138 | 58.6 | 89.79 |
| *ube2* | TsM_001232600 | TTGTCCCACTTATCCCAAGG/ ACGGCCTAAGCACACTGAAC | 100 | 59.1 | 110.49 |
| *mapk3* | TsM_000447200 | TCAAGCCGGACTACTTACCG/ CACGGTTGCTGTACATCTCG | 140 | 58.6 | 101.56 |
| *h2b* | TsM_000989300 | GCTCCTAAAGTAGTGTCAGGCA/ ATGGCGTAGCTCTCCTTCCT | 104 | 63.2 | 100.38 |
| *wnt11a* | TsM_000806100 | GGCAGGCGAGGTTCTGTAAT/ GGCACAGTAGCGACCGATTT | 239 | 60.2 | 101.85 |

compared to cysts in media without TA (p = 0.002) (Fig 2). This supports the use of TA to induce scolex evagination in our *in vitro* model.

**RT-qPCR amplification efficiency and product specificity**

Primers for the 12 *T. solium* sequences of candidate RGs and 2 target genes (Table 2) were first evaluated for specificity using end-point PCR, followed by agarose gel electrophoresis. Each reaction produced a single-band amplicon consistent with the expected product size, demonstrating high primer specificity (Fig 3). These primers were subsequently utilized for qPCR amplification of cDNA templates. The amplification efficiencies ranged from 89.79% to 110.49%, with 12 of the 14 sets falling within the accepted range of 90–110%. These values indicate robust amplification performance across most primer sets, with minor deviations observed in two cases. (Table 2). All average thresholds for Cq determinations consistently fell within the linear segment of amplification curves, indicating reliable signal detection. In addition, the negative first derivative of each melting curve showed a single peak, corroborating strong primer specificity and the absence of non-specific amplification (Fig 4). To evaluate potential gDNA contamination, RT-qPCR reactions with and without reverse transcriptase (+RT/–RT) of representative genes were performed (S1 Fig). Amplicons detectable in –RT controls confirmed the presence of trace gDNA despite DNase treatment. However, mean Cq values in –RT reactions [29–33] were consistently 7–14 cycles higher than in +RT reactions [19–26] (S2 Table), corresponding to <1% contribution to total signal when calculated using the comparative ΔCq method [43]. According to established criteria [44,45], such differences indicate negligible contamination.

**RNA-seq and expression of the candidate RGs**

The average transcript expression, quantified as $\log_2(\mathrm{CPM}+1)$, for all candidate RGs (not including the mitochondrial transcript *cox1*) across the three early growth phases PRE, EV, POST, and both inducer conditions, TA− and TA+, is

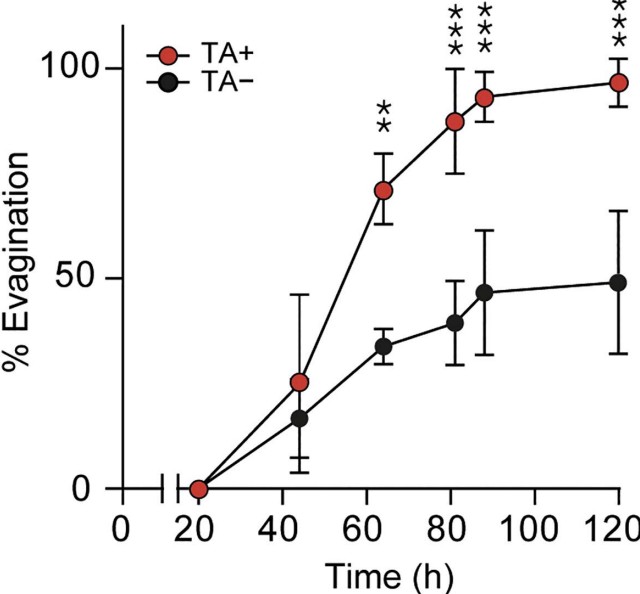

**Fig 2. Taurocholic acid (TA) induces *in vitro* evagination of *T. solium* cysts.** Percentage of evaginated parasites was quantified at the indicated time points under TA− (black circles) or TA+ (red circles) conditions. Each data point represents the mean ± SEM from three independent experiments. *p* < 0.01 (**), *p* < 0.001 (***) indicate significant differences between TA− and TA+ at the corresponding time points.

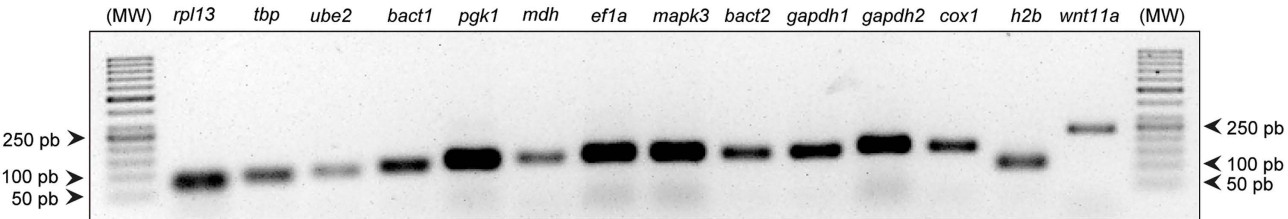

**Fig 3. Primer pair specificity for candidate reference genes.** End-point PCR amplification products for all candidate reference genes (RGs) were resolved by agarose gel electrophoresis to assess primer specificity. Each lane corresponds to a distinct RG, with PCR products arranged according to expected amplicon size. A single band of the anticipated size was observed for each gene, indicating specific amplification. MW: 50 bp DNA molecular weight ladder, with size markers indicated on the left and right.

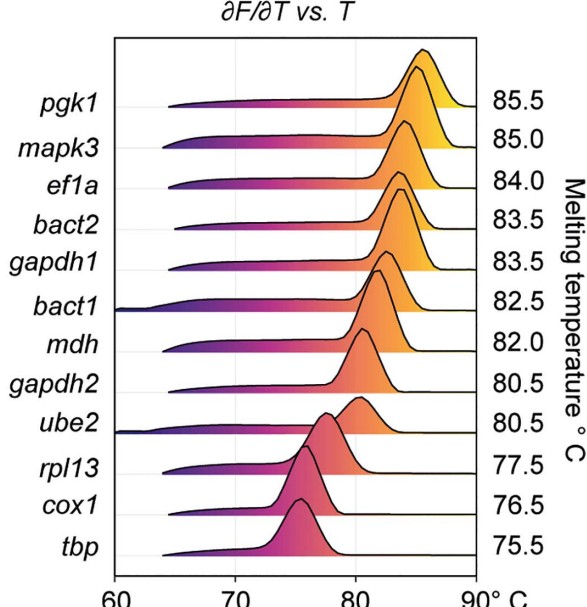

**Fig 4. Ridgeline plot of melting curves for candidate reference genes.** Simplified melting profiles (dF/dT vs. temperature) for each candidate reference gene (RG) derived from qPCR runs are shown as ridgeline plots. Each peak represents the melting temperature (Tm) of the corresponding PCR product, indicating amplicon specificity. A single, sharp peak per gene confirms the absence of non-specific amplification or primer-dimer artifacts.

presented in descending order in Table 3. This value consistently showed a standard deviation lower than 0.4, with the exception of *bact2*. Transcripts *ef1a* and *gapdh1* exhibited the highest average expression levels. Conversely, *gapdh2* and *ube2*, at the bottom of the table, were not highly or even moderately expressed, suggesting they are not suitable as candidate RGs in this study (Table 3).

## qPCR of the candidate RGs

qPCR was performed using cDNA from PRE, EV, and POST parasites, each group under both TA+ and TA- conditions. A heatmap of the raw mean Cq values considering all samples individually (n = 10 per group and culture condition), with transcripts placed from lowest to highest Cq (i.e., highest to lowest abundance of transcripts), helped identify samples contributing to variability within their groups. Two outlier samples from the EV TA+ group were excluded from further analyses (Fig 5).

**Table 3. Expression of candidate RGs. Average log$_2$(CPM+1) and standard deviation for the candidate RGs, obtained from RNA-seq results.**

| Transcript | Average log$_2$(CPM+1) | SD |
|---|---|---|
| *ef1α* | 11.86223 | 0.182343 |
| *gapdh1* | 11.343 | 0.353272 |
| *mdh* | 10.97606 | 0.210379 |
| *bact2* | 10.85346 | 1.042606 |
| *bact1* | 9.843404 | 0.304582 |
| *pgk1* | 8.823891 | 0.339484 |
| *rpb2* | 7.858367 | 0.20439 |
| *rpl13* | 6.831909 | 0.157581 |
| *mapk3* | 5.887588 | 0.316387 |
| *tbp* | 5.367023 | 0.213984 |
| *gapdh2* | 1.438938 | 0.268953 |
| *ube2* | 0.327349 | 0.175001 |

Transcripts *ef1a* and *gapdh1* showed the highest expression across all groups and both culture conditions (PRE, EV, POST; TA+, TA-) (Fig 5, top of the heatmap; Fig 6). Mean Cq values for all candidate RGs were below 35 cycles, suggesting that transcript abundances were not underrepresented and could be detected. Transcripts for *ube2* and *gapdh2* had the highest mean Cq values in the 30–33 range, which implies low transcript abundance and could increase stochastic bias in the qPCR quantification (Fig 6). These high Cq values reinforce the RNA-seq findings, further supporting the exclusion of *ube2* and *gapdh2* as suitable RGs.

### Comprehensive stability ranking of candidate RGs among conditions from RNA-seq

The stability of candidate RGs was assessed using the geometric mean of their RNA-seq rankings, derived from *geNorm* and *NormFinder* analyses under three conditions: TA- only, TA+ only, and TA- and TA+ combined (all) (Fig 7, left side). For

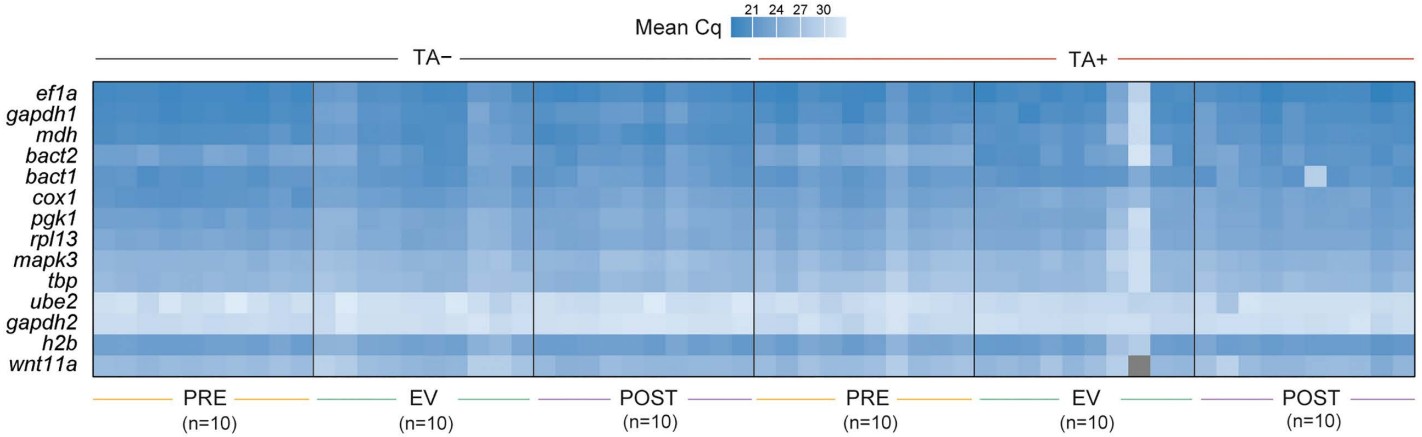

**Fig 5. Raw mean Cq values across parasite samples for candidate reference genes.** Heatmap representation of quantification cycle (Cq) values obtained from qPCR amplification of candidate reference genes (RGs) across all parasite samples. Samples are grouped by treatment condition (TA− and TA+), and time point (PRE, EV, POST; *n* = 10 per group). Darker blue shades indicate lower Cq values (higher expression), while lighter shades indicate higher Cq values (lower expression). Each row corresponds to one gene, and each column to an individual cyst.

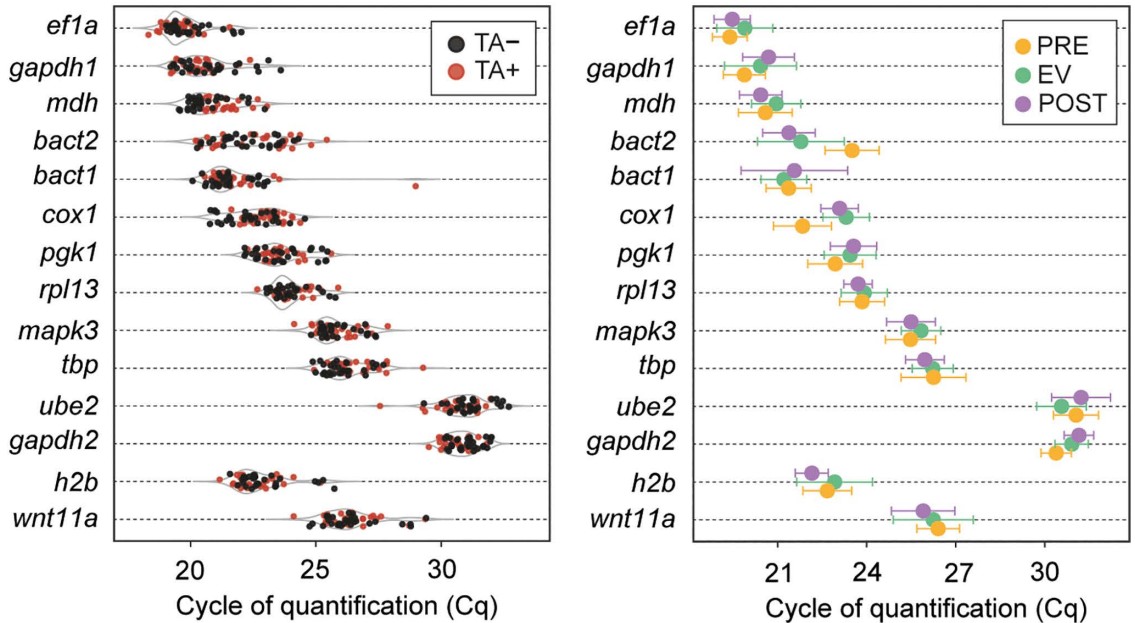

**Fig 6. Distribution of Cq values for candidate reference and target genes under different experimental conditions.** Left panel: Dot plot showing individual Cq values for each gene grouped by treatment condition (TA−: black; TA+: red). Right panel: Mean±SD Cq values grouped by parasite growth phase (PRE: orange; EV: green; POST: purple).

the TA- only, *rpl13*, *mdh*, *bact1* and *ef1α* ranked as the most stable, while *pgk1* and *bact2* were among the least stable candidates. In the TA+ condition, transcripts for *rpl13*, *bact1*, *ef1α* and *mdh* were the top-ranked, whereas *gapdh2* and *bact2* showed poor stability. Finally, in the combined dataset, *rpl13*, *ef1α*, *mdh,* and *bact1* exhibited the lowest geometric means, indicating their high expression stability. Conversely, *gapdh2* and *bact2* showed the highest values, consistently suggesting their poor suitability as RGs.

## Comprehensive stability ranking of candidate RGs among conditions using Cq values

Consistently across all datasets, *rpl13* ranked among the most stable genes. In the combined dataset (TA+ and TA-, all conditions), *rpl13*, *mapk3*, *pgk1* and *ef1α* showed the lowest geometric means, indicating high expression stability, while *bactin1* and *bactin2* were the worst ranked. Similar trends were observed when analyzing TA- samples separately, with *rpl13*, *ef1α*, and *mapk3* maintaining high stability, whereas *cox1*, and *bactin2* were among the least stable. In the TA+ dataset, *pgk1*, *rpl13*, *mdh* and *ef1α* were the top-ranked, while *bactin1* and *bactin2* displayed the highest geometric means (Fig 7, right side).

(S3 Table) contains all stability values for each algorithm, both from RNA-seq and qPCR data.

## Validation of the top-ranked RGs

To validate the reliability of the most stable RGs, *h2b* and *wnt11a* were used as target genes and were compared with their corresponding expression profiles in the RNA-seq dataset. Their relative expression was analyzed across PRE, EV, and POST groups, under both TA+ and TA- conditions, with the PRE group serving as the control. In Fig 8A, relative quantification of *h2b* and *wnt11a* based on normalization with each RG, using the standard 2$^{-\Delta\Delta Ct}$ protocol, is shown.

Fig 9 shows the normalization with the geometrical mean of *rpl13* and *ef1a*, the RGs stably expressed in all conditions (TA+ and TA-, TA+, TA-), considering amplification efficiencies, according to the Pfaffl method (S4 Table). *wnt11a* uniquely delimitates a transition zone between the neck and strobila in the *Hymenolepis* adult stage, mainly associated with

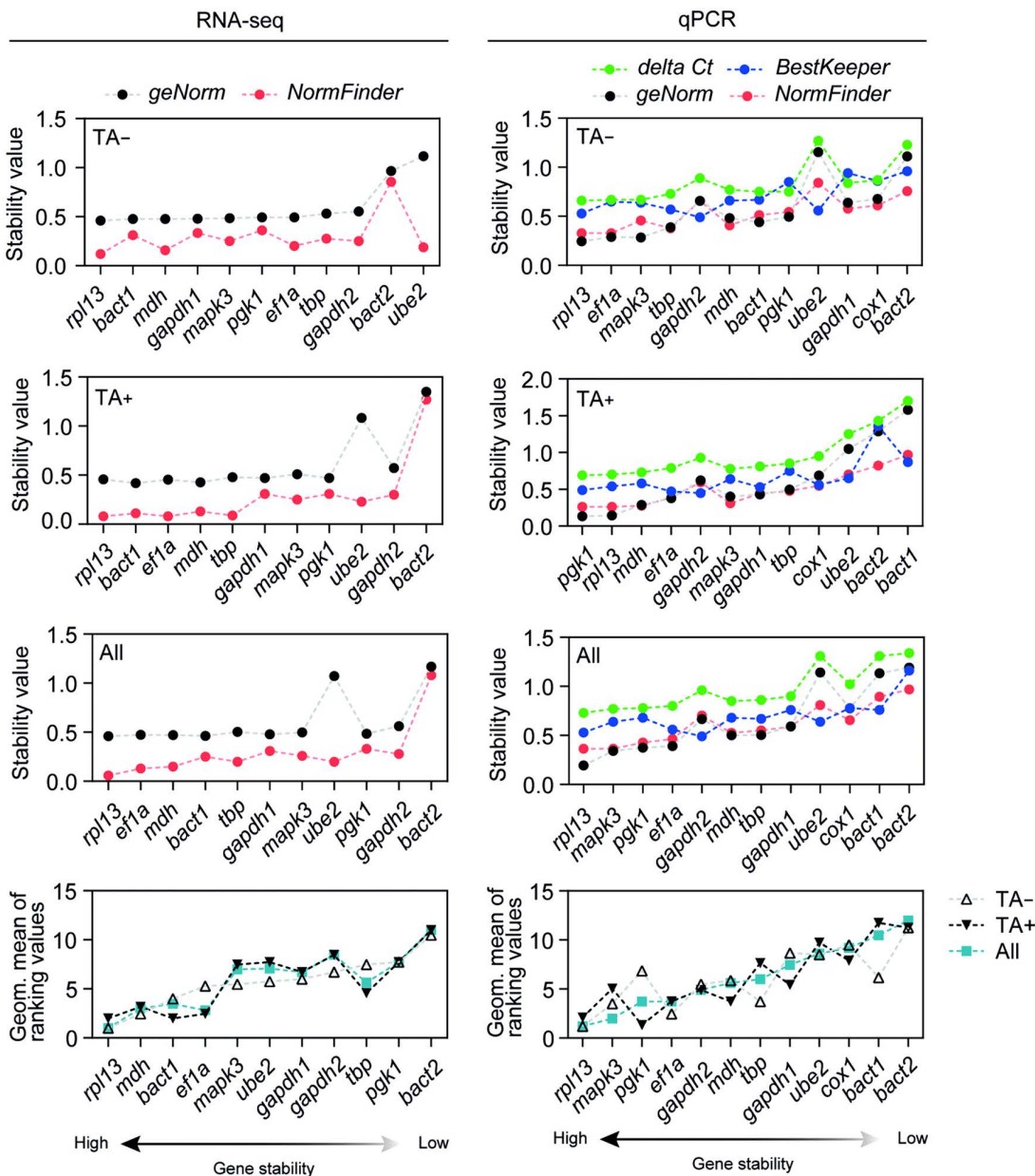

**Fig 7. Comprehensive stability ranking of candidate reference genes based on RNA-seq and qPCR analyses.** Stability values of candidate reference genes (RGs) were assessed using multiple algorithms. Left panels: Stability scores and rankings calculated from RNA-seq-derived expression values using *geNorm* (black) and *NormFinder* (red), along with their geometric mean ranking values across TA−, TA+, and all conditions. Right panels: Stability scores calculated from qPCR Cq values using *delta Ct* (green), *BestKeeper* (blue), *geNorm* (black), and *NormFinder* (red), followed by a comprehensive ranking based on the geometric mean of individual ranks. Lower stability values indicate more stable gene expression. Genes are ordered from most to least stable (left to right) based on the comprehensive ranking.

strobilation [41,42]. We expected this gene to behave similarly comparing PRE and EV, and might appear upregulated in POST. Related to both cell proliferation and strobilation inherent to this morphogenic process, *h2b* expression is expected to show some difference between our chosen developmental phases, as cellular proliferation is required for larval metamorphosis in *Hymenolepis* [22].

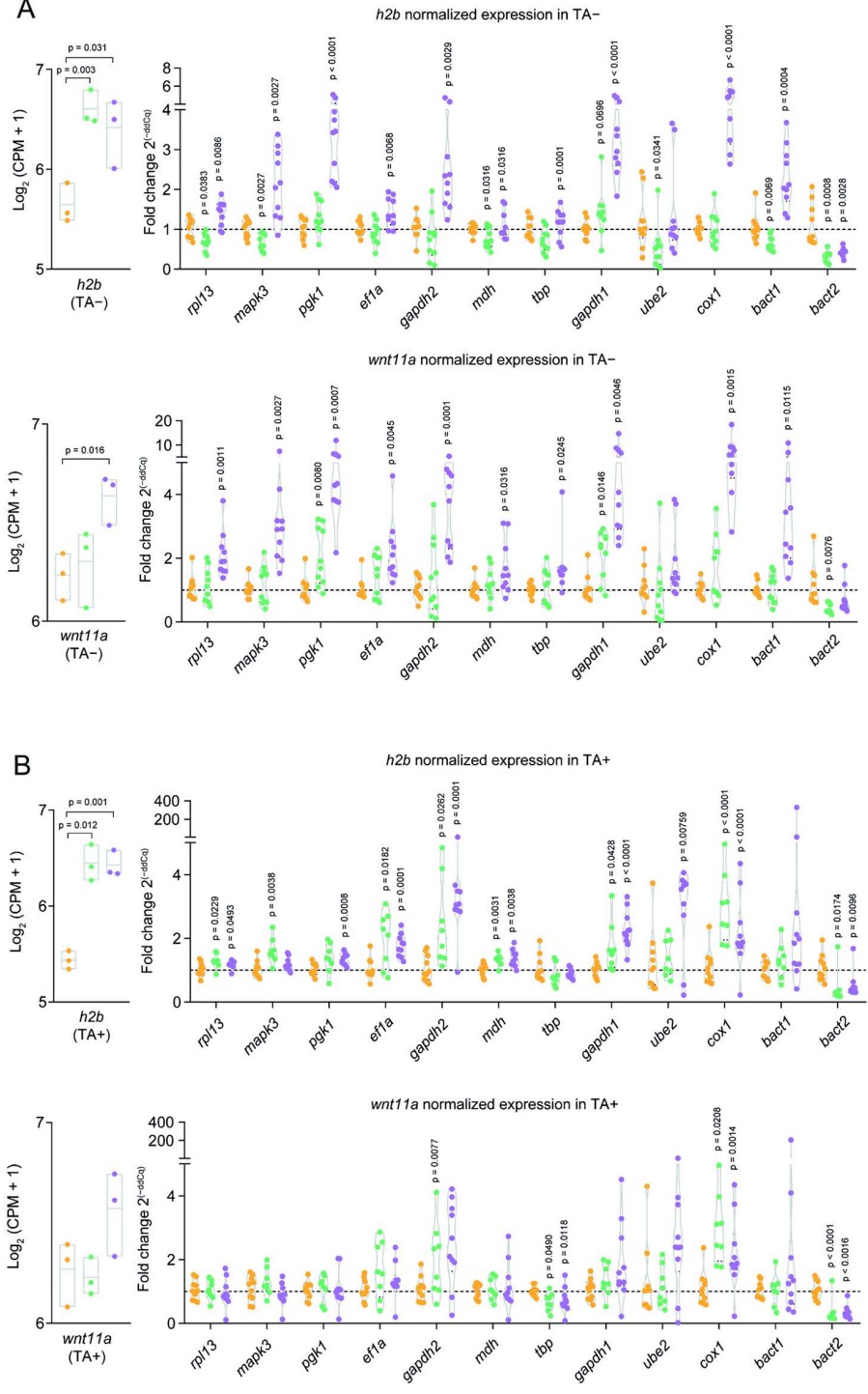

**Fig 8. Relative quantification of *h2b* and *wnt11a* expression normalized to candidate reference genes.** Expression levels of *h2b* (top panels) and *wnt11a* (bottom panels) were evaluated under TA− (A) and TA+ (B) conditions. The left insets display raw expression profiles (log₂(CPM + 1)) based on RNA-seq data, highlighting the expected trends in gene expression across developmental stages (PRE: orange; EV: green; POST: purple). The main panels show relative expression levels calculated using the $2^{-\Delta\Delta Cq}$ method, where each gene's expression was normalized against individual candidate reference genes (RGs). Reference genes are ordered from most to least stable based on prior stability assessments. Each dot corresponds to a single parasite cyst, with values obtained from triplicate qPCR reactions. Statistical comparisons between developmental stages were performed using Student's *t*-test, and the *p*-values for pairwise comparisons are displayed above each respective comparison.

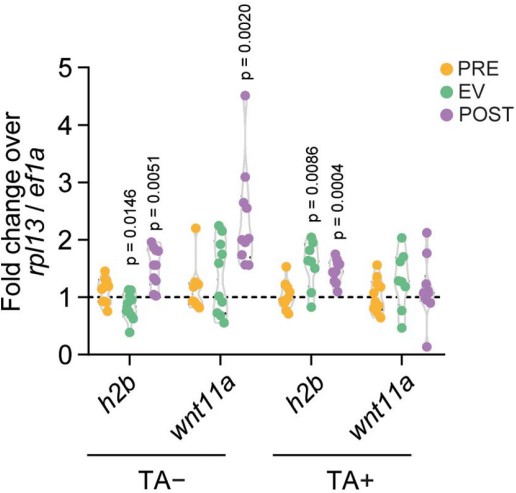

**Fig 9. Relative quantification of *h2b* and *wnt11a* expression normalized to the geometric mean of *rpl13* and *ef1α*.** Fold changes in expression of *h2b* and *wnt11a* were calculated using the Pfaffl method [39], with normalization performed using the geometric mean of *rpl13* and *ef1α*, the top-ranked reference genes. Target genes were measured across developmental stages (PRE: orange; EV: green; POST: purple) under both TA− and TA+ conditions. Each dot represents one parasite cyst, with Cq values derived from triplicate qPCR reactions. Statistical comparisons between stages were conducted using Student's *t*-test, and *p*-values are indicated above each comparison.

Relative quantification of *h2b* and *wnt11a* based on normalization against each candidate RG, starting with the most stable ones, allowed to identify that *rpl13* and *ef1α* are the best to reflect the expected expression pattern of *h2b* and *wnt11a* in TA+ or TA-.

The expression patterns of *h2b* observed in RNA-seq were matched by qPCR analysis only in TA+ (Fig 8). In TA+, qPCR showed that *wnt11a* levels remained constant across development phases, consistent with RNA-seq analysis, although a non-significant tendency to increase its expression is seen towards the later point. Conversely, in TA- conditions, *wnt11a* is overexpressed in more advanced stages after evagination (POST) (Fig 8), which suggest a shift to TA-independent developmental processes.

To ensure the reliability of the selected RGs, expression levels of *h2b* and *wnt11a* were normalized using the geometric mean of *rpl13* and *ef1α*, and also using each RG individually to confirm consistency and rule out bias from either one. Relative quantification of *h2b,* normalized using the geometric mean of *rpl13* and *ef1α* (Fig 9)*,* showed that expression trends remain similar to *h2b* expression when normalized using *rpl13* or *ef1α* separately (Fig 8). In *wnt11a*, the expression patterns also remained similar, with qPCR and RNA-seq congruent results in TA+ and TA-.

## Discussion

The central finding of this study is the identification and validation of stable endogenous reference genes (RGs) essential for accurate single-gene expression analysis during the cyst-to-early adult development of *Taenia solium*. This part of the parasite's lifecycle is crucial for sexual maturation and, therefore, the spread of the parasite; however, it remains largely unexplored, despite its relevance to public health in low-resource regions. Essential molecular tools like this one should enable valuable studies on gene function and regulation under well-defined biological conditions.

The most stable RGs among the six experimental groups (PRE, EV, POST, each TA+ and TA-) were expected to show similar results for the target transcripts *h2b* and *wnt11a* in both the analysis of RNAseq data based on $\log_2(CPM+1)$ and RT-qPCR data.

Among 12 candidates, *rpl13*, *ef1α* and *mdh*, were initially ranked as the most stable for normalizing transcript expression with the RNA-seq data. This was consistently reinforced, as *rpl13* and *ef1α* ranked among the top four most stable transcripts in RT-qPCR analyses across all experimental conditions (three morphologically identified developmental phases (PRE, EV, POST) around the *in vitro* evagination of the scolex, each without or with 0.1% taurocholic acid, TA) (Fig 7).

The geometric mean of the two identified stable RGs (*rpl13*, *ef1α*) was used to normalize qPCR data from two target genes (Fig 9); *h2b* showed the expected expression associated with proliferation in TA+ condition, and *wnt11a* in TA+ and TA-, aligning with biological expectations (Figs 8 and 9). While this element of the Wnt signaling pathway is known for its role in defining the anteroposterior axis in platyhelminths [41], the use of whole organisms possibly masked its upregulation during the phases studied, as it would only be expressed in specific parts of the parasite, mainly in adult stages.

The validation of the reliability of the RGs was crucial. For contrast, normalizing gene expression with less stable candidates (like *bact2*, *ube2*, and *gapdh1*) resulted in significantly variable and inconsistent fold-changes for target genes *h2b* and *wnt11a*, sometimes fluctuating wildly from 10- to 6000-fold (Fig 8). In *Echinococcus*, *bact2* has been shown to be significantly upregulated across multiple developmental stages, confirming it as unsuitable for normalization [46]. This highlights that using unstable genes can lead to misleading conclusions, reinforcing the necessity of RG validation for solid results.

Based on this demonstration, we explicitly state that using the geometric mean of two validated RGs (*rpl13* and *ef1α*) is highly advisable. While three controls are often recommended [32], rigorous analysis confirmed that only these two maintained the necessary unconditional stability required for reliable results in this system, as other stable candidates were excluded due to potential TA-dependent modulation (*mapk3*) or media bias (*pgk1*). This gold-standard practice minimizes bias and improves the reliability of qPCR results by ensuring that results are not driven solely by normalization against a single, potentially unstable, reference gene.

Additionally, *ef1α*, ranked among the top four RGs under all tested conditions (TA+ and TA-, TA+, TA-; Fig 7), has been used to normalize gene expression of Antigen B family genes during the early pre-adult development in E*chinococcus spp* metacestode, and was identified, alongside *tbp,* as one of the most stable RGs in *Echinococcus ortleppi* [21]. However, in a recent transcriptome-based study of the free-living echiuran annelid *Urechis unicinctus*, *tbp* was found to be among the least stable genes, raising questions about its suitability as a housekeeping gene [47]. In our analysis, *tbp* did not rank among either the most or least stable reference genes (Fig 7). The consistent stability of *rpl13* and *ef1α* in *Echinococcus* species, contrasted with the variability of *tbp*, suggests that the expression profiles of commonly presumed housekeeping genes may not be universally conserved—even within the phylum Platyhelminthes.

*gapdh* is commonly used to normalize gene expression, but it plays several roles beyond its well-known function in the glycolytic pathway. For instance, it helps maintain cellular iron homeostasis [48] and acts as a chaperone [49]. Furthermore, its expression levels in human cells are not stable [50] and show greater variability across different tissues compared to other genes [51,52]. Consequently, some studies have raised concerns about using *gapdh* as a RG in qPCR experiments [53].

Moreover, *gapdh* has undergone gene duplication events throughout evolution, resulting in different isoforms that may have distinct roles. As a result, the expression of *gapdh* isoforms can vary depending on the tissue [54]. Given the lack of isoform annotations in the currently available *T. solium* genome, our comprehensive analysis included two *gapdh* paralogs, *gapdh1* and *gapdh2,* but neither demonstrated high stability. Although the *BestKeeper* algorithm identified *gapdh2* as the most stable candidate, likely due to its low SD across all samples and conditions, other algorithms ranked *gapdh2* lower (Fig 7). Additionally, *gapdh2* exhibited the highest raw Cq values (Figs 5 and 6) among the twelve initial RGs, suggesting low expression levels in parasite tissues, making it unsuitable for relative expression analyses. Notably, when *gapdh1* was used to normalize the expression of *h2b* and *wnt11a,* the resulting fold change values were significantly over-estimated compared to those obtained with the top-ranked RGs (Fig 8).

## The relevance of TA for *T. solium* cyst-to-early adult development

The controlled *in vitro* activation of cysts in the presence of TA provides a valuable model for studying *T. solium* early-adult development. The normalization of expression data often relies on canonical RG that have not been properly validated for this organism or its (yet not described) developmental stages. Alongside the identification and validation of suitable RGs, our study provides some biological insights into *T. solium* development, particularly concerning the essential process of its transformation from larval cyst to adult tapeworm. These could be confirmed through new RT-qPCR studies for specific genes, tissues, and events.

The TA+ condition aims to mimic the environment of the small intestine where *Taenia solium* resides in the presence of biliary salts. Bile from different sources, as well as taurocholic acid, has been used in the past as an inducer of *in vitro* scolex evagination [55], but a systematic demonstration of this function is not readily available. In fact, the molecular mechanisms by which TA induces scolex evagination (Fig 2) remain to be explored. We observed an effect of TA on the expression of specific genes, such as *h2b* and *wnt11a*. Distinct expression patterns in TA+ versus TA- conditions suggest potential TA-independent developmental processes (Figs 8 and 9).

The consistent stability of *rpl13* (ribosomal protein L13) and *ef1α* (elongation factor 1 alpha) under both TA+ and TA− conditions (Fig 7) underscores their reliability as reference genes for normalization in these biologically relevant contexts of parasite activation. It has been previously shown that ribosomal proteins are overrepresented in adult *T. solium* [56]. Their sustained expression across the defined phases of the study suggests that components of the translational machinery are constitutively active during early adult development, reflecting essential cellular functions maintained regardless of external stimuli. On the other hand, while the stability of these two main RGs appears unaffected by TA exposure, *mdh* (malate dehydrogenase) exhibited high stability in TA+ samples (Fig 7). As this enzyme is a key component of malate dismutation, a metabolic pathway characteristic of adult cestodes [57], its increased stability in TA+ samples may reflect the activation or maintenance of adult-like metabolic pathways. This supports the idea that TA+ conditions promote or sustain *in vitro* development toward the adult stage.

Although the reference genes *rpl13* and *ef1α* remained stable across all developmental phases and regardless of the presence of TA, their ability to validate biological results did depend on this condition. The normalization of *h2b* expression using their geometric mean aligned with RNA-seq data only under TA+ conditions. In contrast, *wnt11a* expression was consistent with RNA-seq in both TA+ and TA−. This suggests that these reference genes more accurately capture biological variability under TA+ conditions, which better mimic the host environment. The concordance of *h2b* and *wnt11a* expression with RNA-seq data in TA+ cultures may reflect a more coordinated transcriptional program induced by TA as a synchronizing factor. This contrasts with the apparently asynchronous expression pattern observed in TA− cultures. Thus, while the cultures can be sustained in the absence of TA, its inclusion is essential for capturing the coordinated transcriptional dynamics associated with the adult transition and for enhancing the biological relevance of the assays.

Although *mapk3* (mitogen-activated protein kinase 3) ranked among the top four most stable RGs across all conditions (TA+ and TA−) (Fig 7), we do not recommend its use due to evidence of TA-dependent modulation. Specifically, *mapk3* expression appears responsive to TA exposure, which compromises its reliability for normalization. Supporting this, the MAPK signaling pathway has been shown to be differentially active in the hyperproliferative racemose form of the cyst compared to the non-evaginated larval cyst, indicating context-dependent regulation [58].

*pgk1* codes for a glycolytic enzyme and is frequently selected as a housekeeping gene for normalization in qPCR and gene expression analyses in cell lines or animal tissues [59]. However, it has been reported that high glucose exposure upregulates the mRNA levels of PGK1 in human proximal tubular cells [60]. No studies at a molecular level have been performed in free-living or parasitic worms to ensure that absence of glucose or gradients of glucose exposure had any effect on *pgk1* expression. Given that the RPMI medium used in this study is enriched with a standard glucose concentration, we recommend taking the glucose factor into consideration when selecting this gene as a reference.

Together, this study offers validated reference genes and important biological insights into *Taenia solium*'s developmental transition from larval cyst to adult tapeworm—a key phase for transmission within the human intestine.

## Methodological rigor

Our design employed a robust dual approach, integrating both RNA-seq and RT-qPCR data, and various algorithms, to assess RG stability across the three determined groups and two culture conditions. RG stability was evaluated by both a previously generated RNA-seq transcriptome dataset and newly performed two-step RT-qPCR, enabling both *in silico* and experimental validation. The amplification efficiency for most primers designed for the RGs was between 90–110%, and product specificity was confirmed by single-band amplicons and melting curves, adhering to MIQE guidelines. MIQE 2.0 [61] guidelines call for disclosure of primer location and validation data (S1 Table, S1 Fig). Although most primer pairs in our study were located within single exons, gDNA contamination had no meaningful impact on the accuracy of our strategy or the reliability of the expression data. Strict criteria, such as excluding RG datasets with Cq values above 35, were applied to ensure reliable transcript detection and support the exclusion of unsuitable RGs like *ube2* and *gapdh2* due to low abundance and potential stochastic bias.

RNA-seq captured averaged transcriptomic profiles from pooled samples, whereas RT-qPCR analyzed individual cysts to account for biological heterogeneity. RT-qPCR was performed using different RNA samples than those used to generate cDNA for RNA-seq. Using matched samples would have provided a more direct and reliable basis for comparison, helping to prevent the differences that reflect the inherent biological variability introduced when experiments are replicated independently. However, using independent biological samples for each technical assessment provided a successful validation of the reliability and robustness of the selected RGs that accounts for natural variations across different parasite batches.

The initial stability analysis was conducted *in silico* using RNA-seq data derived from 72 parasites, while the subsequent experimental validation involved RT-qPCR performed on additional sixty parasites obtained from a separate culture. This complementary design provides a strong test against the inherent natural variability encountered when experiments are replicated.

Despite the necessary biological separation between the RNA-seq and RT-qPCR samples, the candidates *rpl13* and *ef1a* consistently emerged as the most stable transcripts across all analytical conditions. This strong concordance between two distinct experiments and sampling strategies provides a higher standard of proof for the reliability and suitability of *rpl13* and *ef1a* for future molecular studies of *T. solium* development.

A significant limitation is that our determinations, both for RNA-seq and qPCR, were done on whole-cyst RNA preparations. As seen in other cestodes, several genes exhibit discrete expression patterns along the adult worm's body. Techniques such as *in situ* hybridization have revealed the presence of regional expression gradients, which may be restricted to or extend across anatomical regions like the scolex, the "transition zone" between the neck and the nascent strobila, as well as the immature and mature proglottids [41,42]. Therefore, depending on the specific goals, a more refined set of criteria should be established, such as preparing RNA from carefully selected anatomical regions, to improve the accuracy of the analysis. Nevertheless, this would be difficult on cysts with invaginated scolices (our PRE parasites).

## Supporting information

**S1 Table. InterPro Summary and primer location.** First tab: Gene names, Wormbase accession numbers and descriptions, and InterPro IDs and descriptions for each of the 12 candidate RGs, *h2b,* and *wnt11a*. Second tab: Genomic location of primers across target gene exons.
(XLSX)

**S1 Fig. Evaluation of genomic DNA contamination by RT-PCR reactions with and without reverse transcriptase (+RT/–RT).** Agarose gel electrophoresis of end-point RT–PCR products for *rpl13* (74 bp), *ef1a* (138 bp), *wnt11a* (239 bp), *h2b* (104 bp), *gapdh1* (147 bp), and *mdh* (131 bp). For each target, three reaction conditions are shown: no-sample control

(NS), reaction performed without reverse transcriptase (−RT), and reaction performed with reverse transcriptase (+RT). Molecular weight markers (MW) are shown on both sides of the gel, with fragment sizes indicated in base pairs (bp). (TIF)

**S2 Table. Estimation of genomic DNA contribution to RT-qPCR signal using −RT controls.** Quantitative PCR (qPCR) results obtained from reactions performed in the presence (+RT) or absence (−RT) of reverse transcriptase for 6 target genes. For +RT and −RT conditions, Cq values from three technical replicates (Cq1–Cq3) are shown along with the mean ± SD. The ΔCq value corresponds to the difference between −RT and +RT mean Cq values (ΔCq=(−RT) − (+RT)). Relative signal attributable to genomic DNA (gDNA) contamination was estimated as $2^{(-\Delta Cq)}$. The contribution from −RT (%) represents the percentage of amplification signal arising from gDNA. NA indicates reactions with no detectable amplification. (XLSX)

**S3 Table. Input data for Fig 7.** Stability rankings of the candidate RGs. Separate tabs in the following order: Cq values of all samples (TA+ together with TA-), isolated TA+ Cq values, isolated TA- Cq values, ranking by RefFinder (for qPCR results), CPMs of all samples (TA+ together with TA-), isolated TA+ CPMs, isolated TA- CPMs, $\log_{2\,CPM}$s of all samples (TA+ together with TA-), isolated TA+ $\log_{2\,CPM}$s, isolated TA- $\log_{2\,CPM}$s, ranking by GeNorm-NormFinder of $\log_{2\,CPM}$ values (RNASeq results). (XLSX)

**S4 Table. Data for Fig 9.** Relative quantification of *h2b* and *wnt11a* expression. Separate tabs for Cq Raw Data, Cq means, and Pfaffl results for *h2b* and *wnt11a* in TA+, TA-, and merged results. (XLSX)

## Acknowledgments

We thank Luz Moyano and Ricardo Gamboa (Global Health Tumbes, UPCH) for providing viable *T. solium* cysts; David Durand (Instituto de Medicina Tropical "Alexander von Humboldt", UPCH) for his kind assistance with gene expression equipment; and Francisco Villafuerte, Daniela Bermúdez, and Rómulo J. Figueroa (Laboratorio de Fisiología Comparada, UPCH) for sharing their facilities for parasite culture.

## Author contributions

**Conceptualization:** Renzo Gutierrez-Loli, Juan Blume-La-Torre.

**Formal analysis:** Jose Maravi, David Castaneda-Carpio, Renzo Gutierrez-Loli, Segundo W Del Aguila.

**Funding acquisition:** Renzo Gutierrez-Loli, Cristina Guerra-Giraldez.

**Investigation:** Jose Maravi, David Castaneda-Carpio, Renzo Gutierrez-Loli, Segundo W Del Aguila, Valeria Villar-Davila, Juan Blume-La-Torre.

**Methodology:** David Castaneda-Carpio, Renzo Gutierrez-Loli, Juan Blume-La-Torre.

**Project administration:** Renzo Gutierrez-Loli, Cristina Guerra-Giraldez.

**Resources:** Renzo Gutierrez-Loli, Cristina Guerra-Giraldez.

**Visualization:** David Castaneda-Carpio, Renzo Gutierrez-Loli.

**Writing – original draft:** Renzo Gutierrez-Loli.

**Writing – review & editing:** Jose Maravi, David Castaneda-Carpio, Renzo Gutierrez-Loli, Valeria Villar-Davila, Cristina Guerra-Giraldez.

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
