## [Decision Letter · Decision Letter 0]

24 Sep 2025

Validation of reference genes for RT-qPCR relative expression analysis during cyst-to- early adult

development of *Taenia solium*

Dear Dr. Guerra-Giraldez,

Thank you for submitting your manuscript to PLOS Neglected Tropical Diseases. After careful consideration, we feel that it has merit but does not fully meet PLOS Neglected Tropical Diseases's publication criteria as it currently stands. Therefore, we invite you to submit a revised version of the manuscript that addresses the points raised during the review process.

Please submit your revised manuscript within 60 days Nov 23 2025 11:59PM. If you will need more time than this to complete your revisions, please reply to this message or contact the journal office at plosntds@plos.org. Please include the following items when submitting your revised manuscript:

We look forward to receiving your revised manuscript.

Kind regards,

Pablo Smircich

Academic Editor

Peter Fischer

Section Editor

Shaden Kamhawi

co-Editor-in-Chief

Paul Brindley

co-Editor-in-Chief

**Additional Editor Comments:**

Please address all the concerns raised by the reviewers.

**Journal Requirements:**

At this stage, the following Authors/Authors require contributions: Jose Maravi, David Castaneda-Carpio, Renzo Gutierrez-Loli, Segundo W Del Aguila, Valeria Villar-Davila, Juan Blume La Torre, and Cristina Guerra-Giraldez. Please ensure that the full contributions of each author are acknowledged in the "Add/Edit/Remove Authors" section of our submission form.

- ® on pages: 8, and 9.

5) We have noticed that you have uploaded Supporting Information files, but you have not included a list of legends. Please add a full list of legends for your Supporting Information files after the references list.

Potential Copyright Issues:

i) Figure 1. Please confirm whether you drew the images / clip-art within the figure panels by hand. If you did not draw the images, please provide (a) a link to the source of the images or icons and their license / terms of use; or (b) written permission from the copyright holder to publish the images or icons under our CC BY 4.0 license. Alternatively, you may replace the images with open source alternatives. See these open source resources you may use to replace images / clip-art:

**Reviewers' Comments:**

Reviewer's Responses to Questions

**Key Review Criteria Required for Acceptance?**

**Methods**

-Are the objectives of the study clearly articulated with a clear testable hypothesis stated?

-Is the study design appropriate to address the stated objectives?

-Is the population clearly described and appropriate for the hypothesis being tested?

-Is the sample size sufficient to ensure adequate power to address the hypothesis being tested?

-Were correct statistical analysis used to support conclusions?

-Are there concerns about ethical or regulatory requirements being met?

Reviewer #1: This study uses rigorous and comprehensive methods to determine suitable reference genes (RGs) for reverse transcriptase quantitative PCR. The authors use appropriate number of samples as well as technical and biological replicates. Statistical analysis was clearly stated and described in the methods, legends, and text. I highly commend the authors for their rigor.

Reviewer #2: Regarding the methods, I believe they are well designed and described with sufficient technical detail. However, one point that raises a question is the normalization strategy: the authors normalized the RNA-seq data using DESeq2 and then calculated CPM values with edgeR. It is not typical to use normalization results from one tool, which relies on specific assumptions, and then derive CPM values with another tool based on that prior normalization.

If this approach is indeed valid, I would suggest providing a clearer explanation of the rationale and methodological details for this step.

**Results**

-Does the analysis presented match the analysis plan?

-Are the results clearly and completely presented?

-Are the figures (Tables, Images) of sufficient quality for clarity?

Reviewer #1: The logical flow for accessing primer stabilities and unvarying expression across many conditions is very well laid out. The authors convincingly show that some common transcripts used of qPCR normalization are less than ideal. Eight primer pairs emerged as potential RGs. To experimentally validate putative RGs, the authors collected 3 stages of T. solium development in the presence or absence of an excystment trigger, taurocholic acid (TA), for which matched RNA-seq data is available. While there were variations in the trends, this is to be expected for qPCR data from complex tissues and timepoints. There was general concordance between qPCR and RNA-seq trends, especially with the preferred RGs. It is unfortunate that there aren’t clearer conditions that the authors can use to biologically validate their RGs. The difference in h2b expression +/- TA is not strikingly different even from the RNAseq data. Expression of wnt11a is even more variable but that could represent real biological variation.

The authors make valid claims that some of the 8 RGs could be differentially expressed in different biologically relevant contexts. They recommend rpl13 and ef1a as RGs. They perform fold change analysis using the combination of those primer pairs as normalization controls as an example of how gold-standard qPCR analysis should be performed. I think this was an important demonstration considering how relying on a single RG can produce results that are driven by normalization. I recommend that the authors strengthen the language discussing this (paragraph at Line 395 or in the discussion). I think it would be valuable for them to explicitly state that using the geometric mean of two endogenous controls is highly advisable.

I have only one concern that needs to be experimentally addressed- potential DNA contamination. The authors state that they DNase treat the extracted RNA but there is no data showing that these primers are not detecting DNA. In my experience, stable qPCR data can be erroneously obtained when there is DNA contamination. Furthermore, DNase from commercial kits are not always effective, especially if the treatment was done in-column. There are no details about primer design (at intron-exon boundaries or flanking large introns) to alleviate these concerns. Figure 4 showing the melting curves for a single product would not detect DNA contamination if the genomic product is the same size (i.e. primers were designed within single exons). The state of the genome may not be reliable enough for such stringent design parameters anyway. The solution is a simple one: perform end-point RT-PCR with and without reverse transcriptase (+/- RT) and show the gel. I would be perfectly satisfied with data showing this for rpl13, ef1a, h2b, and wnt11a only.

Reviewer #2: The results are clearly presented, and the figures are well aligned with and supportive of the findings described in the text.

I believe it is important that the authors address or discuss deeply certain inconsistencies observed. While two strong reference genes, rpl13 and ef1a, are identified, Figure 8A shows that for h2b in TA–, the RT-qPCR results do not replicate what was observed in the RNA-seq data. Moreover, when using rpl13 as a normalizer, a significant downregulation of h2b is detected in EV vs. PRO, whereas the RNA-seq results show the opposite trend. It would be valuable if the authors could explain or provide possible reasons for these discrepancies.

In line with the previous comment, if possible, the inclusion of RT-qPCR results generated from the same RNA samples used for RNA-seq would greatly strengthen the robustness and consistency of the direct comparison between both approaches. I understand this may not be feasible if no RNA remains from the original experiments. In that case, I would suggest acknowledging this point as a limitation of the study, and mention that including RT-qPCR data generated from the same RNA samples used for the RNA-seq experiments would have been particularly valuable. Concerning that this approach would have provided a more direct and reliable basis for comparison, while also helping to disentangle differences that may simply reflect the inherent biological variability introduced when experiments are replicated independently.

**Conclusions**

-Are the conclusions supported by the data presented?

-Are the limitations of analysis clearly described?

-Do the authors discuss how these data can be helpful to advance our understanding of the topic under study?

-Is public health relevance addressed?

Reviewer #1: The conclusions are supported and there are no over-interpretations. The public health relevance was also discussed well.

Reviewer #2: The conclusions are appropriately formulated and consistent with the results obtained.

**Editorial and Data Presentation Modifications?**

Reviewer #1: None

Reviewer #2: (No Response)

**Summary and General Comments**

Reviewer #1: Taenia solium is an important human pathogen ranked by the WHO as a neglected tropical disease though little is known about gene expression changes at development stages of these parasitic flatworms. Ever growing sequencing modalities are expanding our toolkit, but there is a pressing need to perform validations at the bench, and this study is making an important contribution to that effort. This paper validates reference genes (RGs) for use as endogenous controls in qPCR. They use quality control of various primer and amplicon stability programs in addition to qPCR validations compared to RNAseq data. They narrow down a list of acceptable RGs and use them to compare changes at three developmental stages for two markers that indicate increased proliferation and increased strobilation respectively. With a single exception (the lack of data addressing potential DNA contamination), the rigor shown in this paper is excellent and will provide a blueprint for other researchers to assess RGs for their species of interest. The authors also demonstrate a good example of how more reliable qPCR conclusions can be drawn using more than one RG for normalization.

Reviewer #2: I found this work to be both interesting and relevant, particularly in regard to the questions it addresses. The search for suitable reference genes as normalizers for RT-qPCR experiments is highly important, as it directly impacts the accuracy and reliability of gene expression analyses. In this context, it would also be valuable for the authors to discuss certain methodological choices and potential limitations, such as the normalization strategy used for RNA-seq and the possibility of including RT-qPCR data from the same RNA samples. Addressing these points would further strengthen the study and provide a clearer basis for comparison between RNA-seq and RT-qPCR results.

PLOS authors have the option to publish the peer review history of their article (what does this mean? ). If published, this will include your full peer review and any attached files.

**Do you want your identity to be public for this peer review?** For information about this choice, including consent withdrawal, please see our Privacy Policy .

Reviewer #1: No

Reviewer #2: No

**Figure resubmission:**
---

## [Decision Letter · Decision Letter 1]

17 Dec 2025

development of *Taenia solium*Response to ReviewersRevised Manuscript with Track ChangesManuscript

Shaden Kamhawi

co-Editor-in-Chief

Paul Brindley

co-Editor-in-Chief

**Journal Requirements:**

1) Please upload the figures in a correct numerical order in the online submission form.

2) Please also ensure that the supplementary files citations and descriptions match with the file names in the file inventory when uploading them. (e.g. description: S1 Table, file name: S1 Table).

**Reviewers' comments:**

**Key Review Criteria Required for Acceptance?**

**Methods**

-Are the objectives of the study clearly articulated with a clear testable hypothesis stated?

-Is the study design appropriate to address the stated objectives?

-Is the population clearly described and appropriate for the hypothesis being tested?

-Is the sample size sufficient to ensure adequate power to address the hypothesis being tested?

-Were correct statistical analysis used to support conclusions?

-Are there concerns about ethical or regulatory requirements being met?

Reviewer #1: (No Response)

Reviewer #2: (No Response)

**Results**

-Does the analysis presented match the analysis plan?

-Are the results clearly and completely presented?

-Are the figures (Tables, Images) of sufficient quality for clarity?

Reviewer #1: (No Response)

Reviewer #2: (No Response)

**Conclusions**

-Are the conclusions supported by the data presented?

-Are the limitations of analysis clearly described?

-Do the authors discuss how these data can be helpful to advance our understanding of the topic under study?

-Is public health relevance addressed?

Reviewer #1: (No Response)

Reviewer #2: (No Response)

**Editorial and Data Presentation Modifications?**

Reviewer #1: (No Response)

Reviewer #2: (No Response)

**Summary and General Comments**

Reviewer #1: Thank you for addressing all my concerns. The way the data is presented suggests that the authors do not intend to include Table R1, Table R2 and Fig. R1 in the manuscript and that they were meant for reviewers only. Please include them in the supplement. Table R1 can be merged into Table S1 easily. While I agree that the genomic contamination did not significantly alter the findings in this paper, readers may be surprised to discover that Cq values between 29 and 33 can be obtained in the ABSENCE of cDNA. Cq values tend to be lower for RGs than for any given gene of interest. Thus, I believe it is valuable to report this.

Reviewer #2: The authors have critically discussed and adequately addressed the suggestions raised during the review process.

PLOS authors have the option to publish the peer review history of their article (what does this mean? ). If published, this will include your full peer review and any attached files.

**Do you want your identity to be public for this peer review?** For information about this choice, including consent withdrawal, please see our Privacy Policy .

Reviewer #1: No

Reviewer #2: No

**Figure resubmission:****Reproducibility:** To enhance the reproducibility of your results, we recommend that authors of applicable studies deposit laboratory protocols in protocols.io, where a protocol can be assigned its own identifier (DOI) such that it can be cited independently in the future. Additionally, PLOS ONE offers an option to publish peer-reviewed clinical study protocols. Read more information on sharing protocols at https://plos.org/protocols?utm_medium=editorial-email&utm_source=authorletters&utm_campaign=protocols

---

## [Editor Report · Decision Letter 2]

23 Dec 2025

Dear Dr Guerra-Giraldez,

We are pleased to inform you that your manuscript 'Validation of reference genes for RT-qPCR relative expression analysis during cyst-to- early adult development of *Taenia solium* ' has been provisionally accepted for publication in PLOS Neglected Tropical Diseases.

Best regards,

Pablo Smircich

Academic Editor

Peter Fischer

Section Editor

Shaden Kamhawi

co-Editor-in-Chief

Paul Brindley

co-Editor-in-Chief

---

## [Editor Report · Acceptance letter]

Dear Dr Guerra-Giraldez,

We are delighted to inform you that your manuscript, "

Validation of reference genes for RT-qPCR relative expression analysis during cyst-to- early adult

development of *Taenia solium*," has been formally accepted for publication in PLOS Neglected Tropical Diseases.

Best regards,

Shaden Kamhawi

co-Editor-in-Chief

Paul Brindley

co-Editor-in-Chief
